# Pentoxifylline for Renal Protection in Diabetic Kidney Disease. A Model of Old Drugs for New Horizons

**DOI:** 10.3390/jcm8030287

**Published:** 2019-02-27

**Authors:** Javier Donate-Correa, Víctor G. Tagua, Carla Ferri, Ernesto Martín-Núñez, Carolina Hernández-Carballo, Pablo Ureña-Torres, Marta Ruiz-Ortega, Alberto Ortiz, Carmen Mora-Fernández, Juan F. Navarro-González

**Affiliations:** 1Unidad de Investigación, Hospital Universitario Nuestra Señora de Candelaria, 38010 Santa Cruz de Tenerife, Spain; jdonate@ull.edu.es (J.D.-C.); vtagua@funcanis.es (V.G.-T.); carlamferri@gmail.com (C.F.); emarnu87@gmail.com (E.M.-N.); carmenmora.fdez@gmail.com (C.M.-F.); 2GEENDIAB (Grupo Español para el estudio de la Nefropatía Diabética), Sociedad Española de Nefrología, 39008 Santander, Spain; 3Servicio de Medicina Interna, Hospital Nuestra Señora de Candelaria, 38010 Santa Cruz de Tenerife, Spain; carolinahdezcarballo@gmail.com; 4Department of Dialyisis, AURA Nord, Saint Ouen, 93400 Paris, France; urena.pablo@wanadoo.fr; 5Department of Renal Physiology, Necker Hospital, University Paris Descartes, 75006 Paris, France; 6Laboratorio de Biología Celular en Enfermedades Renales, Universidad Autónoma Madrid, IIS-Fundación Jiménez Díaz, 28004 Madrid, Spain; marta.ruiz.ortega@uam.es; 7REDINREN (Red de Investigación Renal-RD16/0009/0007), Instituto de Salud Carlos III, 28029 Madrid, Spain; 8Departamento de Nefrología e Hipertensión, IIS-Fundación Jiménez Díaz y Facultad de Medicina, Universidad Autónoma de Madrid, 28049 Madrid, Spain; aortiz@fjd.es; 9REDINREN (Red de Investigación Renal-RD16/0009/0001), Instituto de Salud Carlos III, 28029 Madrid, Spain; 10REDINREN (Red de Investigación Renal-RD16/0009/0022), Instituto de Salud Carlos III, 28029 Madrid, Spain; 11Servicio de Nefrología, Hospital Universitario Nuestra Señora de Candelaria, 38010 Santa Cruz de Tenerife, Spain; 12Instituto de Tecnologías Biomédicas, Universidad de La Laguna, 38010 Santa Cruz de Tenerife, Spain

**Keywords:** pentoxifylline, diabetic kidney disease, inflammation, Klotho

## Abstract

Diabetic kidney disease is one of the most relevant complications in diabetes mellitus patients, which constitutes the main cause of end-stage renal disease in the western world. Delaying the progression of this pathology requires new strategies that, in addition to the control of traditional risk factors (glycemia and blood pressure), specifically target the primary pathogenic mechanisms. Nowadays, inflammation is recognized as a critical novel pathogenic factor in the development and progression of renal injury in diabetes mellitus. Pentoxifylline is a nonspecific phosphodiesterase inhibitor with rheologic properties clinically used for more than 30 years in the treatment of peripheral vascular disease. In addition, this compound also exerts anti-inflammatory actions. In the context of diabetic kidney disease, pentoxifylline has shown significant antiproteinuric effects and a delay in the loss of estimated glomerular filtration rate, although at the present time there is no definitive evidence regarding renal outcomes. Moreover, recent studies have reported that this drug can be associated with a positive impact on new factors related to kidney health, such as Klotho. The use of pentoxifylline as renoprotective therapy for patients with diabetic kidney disease represents a new example of drug repositioning.

## 1. Diabetes Mellitus and Diabetic Kidney Disease

Diabetes mellitus (DM) represents one of the most important health problems worldwide. Nowadays, more than 450 million people have DM and, according to recent estimations, about 690 million people will present this pathology by 2045 [1]. Subsequently, target organ complications secondary to DM, especially micro- and macro-vascular complications, will constitute one of the most important medical concerns in the near future.

Diabetic kidney disease (DKD) is a relevant complication of DM, constituting the single most common cause of end-stage renal disease (ESRD) in the western world [2]. According to recent estimations, more than 40% of diabetic patients, especially in the case of type 2 diabetes, may develop DKD [3], which generates significant social and economic burdens [4]. In addition, the presence of any stage of DKD is strongly associated with the development of cardiovascular disease, and therefore renal involvement is a major cause of morbidity and mortality in the diabetic population. The pathogenesis of DKD includes mesangial expansion, impairment of endothelial cell function and loss of podocytes in the glomerulus, and interstitial fibrosis in the tubular compartment. The most important clinical manifestation is proteinuria, together with a progressive decline in renal function [5].

Regarding the treatment of DKD, current practice guidelines are focused on halting or delaying the progression of the disease by adequate metabolic regulation and control of the blood pressure (BP), with blockade of the renin-angiotensin aldosterone system (RAAS) as a cornerstone therapy [6]. RAAS blockers such as angiotensin converting enzyme inhibitors (ACEIs) and angiotensin receptor blockers (ARBs) are effective in slowing progression of the disease, but this approximation does not generally halt the progression to ESRD. The combination of RAAS blockers has also been tried but has not been proven to be more effective than monotherapy and is associated with increased adverse events [7,8]. Therefore, there is a need to evaluate new strategies to improve kidney function, delay the progression of the disease, and eventually improve kidney survival. These new approaches become even more necessary if we consider that recent trials designed to find effective renoprotection in DM patients have failed [9,10] or were prematurely stopped because of safety concerns [8,11,12]. Recent studies with new antidiabetic drugs (sodium-glucose co-transporter-2 inhibitors and glucagon-like peptide-1 receptor agonists) have shown beneficial renal effects, and thus, these new drugs have emerged as promising drug classes for treating DKD.

## 2. Old Drug Repositioning

De novo drug discovery is a very costly and time-consuming process. The discovery of one drug takes more than 10 years, with more than one billion USD as the overall estimate of costs and with less than 10% chances of success due to several reasons, including failure rates, high cost, poor safety, poor bioavailability, and limited efficacy. Thus, new strategies for drug discovery have been needed. One of these new approaches is based on the process of finding new uses for existing drugs outside the scope of the original indication; i.e., drug repositioning. This concept evolved in the early 1990s from the fact that different diseases share common molecular pathways and targets in the cell. It gives an extended life for marketed drugs via new indications [13].

There are several examples of successful drug repositioning [13,14,15] (Table 1). Thalidomide was used in pregnant women to prevent morning sickness, but it was withdrawn after cases of phocomelia in newborn babies had been reported. It was repositioned for the treatment of multiple myeloma. Minoxidil, which was initially approved for the treatment of hypertension, was repositioned for the treatment of male pattern baldness, based on the finding that it promotes facial hair growth. Sildenafil is a phosphodiesterase-5 inhibitor that was initially used for the treatment of angina, but it was switched to the treatment of erectile dysfunction. In this context, pentoxifylline (PTX) could be a potential candidate for repositioning based on its beneficial effects in the treatment of DKD.

PTX (3,7-dimethyl-1-(5-oxohexyl)-3,7-dihydro-1H-purine-2,6-dione) is a methyl-xanthine derivative that was approved by the United States Food and Drug Administration for the treatment of intermittent claudication more than 30 years ago [16]. The primary hemorheological effects of PTX are due to increased red blood cell deformability and decreased blood viscosity, although it affects almost all factors responsible for blood viscosity and can be considered as an almost complete rheological drug [17]. However, PTX also has important effects as a modulator of inflammation [18], which supports its use as a renoprotective drug in DKD.

## 3. Inflammation in Diabetic Kidney Disease

The understanding of the pathophysiologic processes leading to DKD has evolved tremendously in recent years. Renal injury was previously explained by metabolic and hemodynamic alterations, which increase systemic and intraglomerular pressure, and by the modification of molecules under hyperglycemic conditions. Nowadays, it is recognized that both the chronic low-grade inflammation and the activation of the innate immune system occurring in DM are related to diabetic complications, becoming key pathophysiological mechanisms involved in the development and progression of DKD.

Plasma concentrations of inflammatory molecules, including proinflammatory cytokines, are elevated in patients with DM [19,20]. Proinflammatory cytokines can lead to the development of microvascular diabetic complications, including nephropathy. Recent studies have shown that the concentrations of these substances increase as nephropathy progresses [21,22] being independently related to clinical markers of glomerular and tubulointerstitial damage, including urinary albumin excretion (UAE), the clinical hallmark of DKD [22,23]. Moreover, accumulation of inflammatory cells in the kidney is closely associated with DKD, and, indeed, inhibition of inflammatory cell recruitment into the kidney has been related to protective effects in experimental models of DKD [24,25,26]. Proinflammatory cytokines synthesized and secreted by these cells in the local microenvironment directly damage kidney architecture and subsequently trigger the epithelial-to-mesenchymal transition process [27], resulting in extracellular matrix accumulation. Furthermore, the expression of chemoattractant cytokines and adhesion molecules is upregulated in kidney cells of diabetic patients. These molecules are key mediators of renal injury by virtue of their ability to attract circulating white blood cells (monocytes, neutrophils, and lymphocytes) and facilitate their transmigration into the renal tissue. These infiltrating cells become a source of new cytokines and other mediators that contribute to the development and progression of renal injury feeding back the process and amplifying the inflammatory reaction.

In diabetic patients, plasma levels of inflammatory cytokines are strong predictors of the development and progression of several renal disorders, including DKD [23,28,29]. Serum and urinary levels of interleukin (IL) 18, a potent proinflammatory cytokine, have been reported to be higher in patients with DKD than in control subjects, showing significant positive correlations with UAE [30]. Many cells produce this cytokine, such as infiltrating monocytes, macrophages, and T cells, and, importantly, tubular renal cells also show increased expression levels of IL18 in patients with DKD [31]. This enhanced expression has been related to the triggering of the mitogen-activated protein kinase (MAPK) pathways secondary to the action of TGF-β [32]. In addition, diverse renal cells (endothelial, epithelial, mesangial, and tubular cells) are also capable of synthesizing other proinflammatory cytokines such as tumor necrosis factor (TNF)α, IL1, and IL6, and therefore these cytokines, acting in a paracrine or autocrine manner, may induce a variety of effects on different renal structures [33,34] playing a significant role in pathophysiology of DKD.

Many clinical studies have reported that the serum and urinary concentrations of TNFα are elevated in patients with DKD as compared with nondiabetic individuals or with diabetic subjects without renal involvement and that these concentrations increase concomitantly with the progression of renal damage [35]. This indicates a potential relationship between the elevated levels of TNFα and the development and progression of renal injury in DM [23,29,36]. This cytokine is cytotoxic to glomerular, mesangial, and epithelial cells and may induce significant renal damage [37]. The direct harmful effect of TNFα on the protein permeability barrier of the glomerulus is independent from alterations in hemodynamic factors or effects of recruited inflammatory cells [38]. In DKD, it is particularly relevant that urinary TNFα has been suggested as a critical factor contributing to sodium retention and renal hypertrophy, important renal alterations that occur during the initial stage of this disease [39]. Moreover, it has been demonstrated that increased urinary as well as renal interstitial concentrations of TNFα precede the rise in albuminuria [40].

Similarly, IL6 levels are also higher in patients with DKD in comparison with diabetic patients without nephropathy [41]. Human renal samples also present increased expression levels of mRNA encoding IL6 in cells infiltrating the mesangium, the interstitium, and the tubules, with a positive correlation with the severity of mesangial expansion [41]. Other functional and structural abnormalities related to DKD and the progression of renal damage have been associated with IL6, including abnormalities in the permeability of glomerular endothelium, the expansion of mesangial cells and enhanced expression of fibronectin [42], the increase in the thickness of the glomerular basement membrane [43,44], and renal hypertrophy [39].

The glomerulus has been the focus of research into DKD, although tubulointerstitial injury is also a major feature, with pathological changes in the tubulointerstitium being closely correlated with renal dysfunction [45]. Accumulating evidence indicates that increased UAE is not simply an aftermath of glomerular injury but is also involved in the development and progression of this complication, since it has been identified as a mechanism associated with the induction of tubulointerstitial inflammation [46].

## 4. Pentoxifylline: Renoprotection and Targeting Inflammation in Diabetic Kidney Disease

PTX is clinically used to treat intermittent claudication resulting from peripheral vascular disease [47]. This drug exerts hemorheological actions, since it reduces blood viscosity, erythrocyte aggregation, erythrocyte rigidity, and platelet aggregation. The improvement in red blood cell flexibility and deformability leads to an improved blood flow [48,49]. This property, together with its potential to decrease intraglomerular pressure, led to an early interest in PTX as a therapeutic agent in kidney disease [50,51]. In fact, data derived from clinical studies and animal models support the use of PTX as an antiproteinuric agent. Interestingly, this antiproteinuric property has been related with its, more recently described, anti-inflammatory effect [52,53,54,55,56,57,58].

PTX is able to modulate TNFα levels by inhibiting the gene transcription and blocking mRNA accumulation [52,59]. Likewise, it has a considerable capacity to modulate other pro-inflammatory cytokines, including IL1, IL6, interferon γ [60,61,62], and other molecules like the intercellular adhesion molecule 1 (ICAM1), the vascular cell adhesion molecule 1 (VCAM1), and the reactive C protein (CRP) [63,64]. Since DKD is a proinflammatory state [44] with increased glomerular permeability to proteins [38], the anti-inflammatory effect of PTX could result in a reduction of proteinuria. Therefore, PTX might represent a novel therapeutic approximation to the treatment of DKD.

To date, a number of clinical trials evaluating PTX in renal patients, most of them diabetic subjects, have been conducted (Table 2).

The first clinical evidence of the renal protective effects of PTX was reported in 1982 by Blagosklonnaia et al. [50]. In that work, administration of 300 mg/day of PTX for three weeks to diabetic patients improved glomerular filtration rate (GFR) and decreased proteinuria. However, it was not until almost the turn of the century that the interest in the anti-proteinuric effects of PTX was renewed. In 1999, Navarro et al. [65] reported, in a group of diabetic patients with advanced renal failure, a decrease both in serum TNFα and proteinuria after treatment with PTX (400 mg/day) for 6 months. In 2005, Aminorroaya et al. [66] and Rodríguez-Morán et al. [67] observed that the administration of 400 mg PTX three times daily to non-hypertensive patients with type 2 diabetes displayed anti-proteinuric effects comparable to those achieved with ACEI treatment. In the same year, Navarro et al. [68], in a randomized, open-label trial, found that an add-on therapy of PTX at a dose of 1200 mg/day for 4 months in DM patients with a background of ARB additively decreased proteinuria. Importantly, this extra antiproteinuric effect of PTX was associated with significant reductions in serum and urinary levels of TNFα, although only variations in urinary TNFα correlated with the change of albuminuria. In a later study in 2006, Rodríguez-Morán et al. [69] also found a reduction in the levels of both high and low molecular weight urinary protein excretion in DM patients with microalbuminuria.

The anti-proteinuric effect of PTX has also been found in non-diabetic subjects. In 2006, Chen et al. [70] reported that the treatment with PTX (800 mg/day for 6 months) decreased proteinuria in 17 patients with primary glomerulonephritis. The reduction of proteinuria was associated with a decline in urinary monocyte chemoattractant protein (MCP) 1 excretion, which allowed the authors to propose a mechanistic basis for PTX in non-diabetic patients with proteinuria. One year later, Shu et al. [71] reported a reduction of proteinuria in non-diabetic patients with chronic allograft nephropathy and microalbuminuria. The Thl/Th2 intracytoplasmic cytokine pattern analysis of peripheral blood CD4+ cells showed a significant decrease of cells bearing TNFα and IL10. Moreover, the graft function was stabilized in more than a half of the patients by the end of the study, pointing to a renal protection exerted by PTX.

The potential renoprotective effect of PTX was posteriorly evaluated in diverse clinical trials, some of which also examined the anti-inflammatory effect of this drug. However, these trials include a variety of study designs, drug dosages, and follow-up periods, which results in inconclusive results. In 2007, Diskin et al. [72], in an open-label, controlled trial including DM patients with nephrotic proteinuria, did not find any additive anti-proteinuric or renoprotective effects of PTX treatment on the background of ACEI and ARB therapy after 1 year of follow-up. However, this study was non-randomized, included only 14 participants, and used dual RAAS blockade, which has been related to important safety concerns [7,8,80]. Another randomized clinical trial published by Badri et al. in 2013 [73] with add-on PTX therapy to background RAAS blockade in a small group of non-diabetic patients showed a reduction of proteinuria in the PTX group without affecting eGFR. In contrast with these results, Perkins et al. [74] observed in 2009 an amelioration of renal function decline in 40 diabetic patients with chronic kidney disease (CKD) stages 3 and 4 after 1 year of add-on PTX to RAAS blockade. However, the authors did not observe a decrease in proteinuria levels and proposed that this parameter does not always constitute an optimal surrogate outcome in these studies. In 2012, Goicoechea et al. [75] reported a stabilization of renal function and a reduction of several markers of inflammation (TNFα, fibrinogen, and high sensitivity CRP) in CKD patients with stage 3 or higher. Again, PTX therapy did not reduce proteinuria in this group, but the high percentages of dropped-out and incomplete follow-up patients should be noted. Similarly, in 2008, Lin et al. found [76] in CKD stage 3 patients with macroalbuminuria that 1 year of treatment with add-on PTX to ARB background therapy reduced proteinuria and urinary levels of TNFα and MCP1 as compared with the ARB monotherapy group. Moreover, further analysis revealed a significant decrease of eGFR in the ARB group but not in the add-on PTX group after 1 year of follow-up.

A few larger studies comprising a higher number of participants have been more conclusive, clearly showing the anti-proteinuric ability of PTX and pointing to the therapeutic benefits of the clinical use of PTX in renal disease. In 2015, Navarro et al. [77] published the PREDIAN trial, to date the largest randomized controlled study to evaluate the renoprotective effects of PTX. The study comprised 169 type 2 diabetic subjects with CKD stages 3 and 4 and residual albuminuria despite RAAS blockade, who were randomized to a control group or an active group. Patients in the active group received PTX (1200 mg/day) on top of RAAS blockers. After 2 years of follow-up, the rate of progression of renal disease was reduced in the PTX group, which was accompanied by a decrease in proteinuria and a reduction in the urinary excretion of TNFα. The smaller decrease of eGFR in the PTX group with respect to the control group showed a trend at 6 months and reached statistical significance after 1 year, suggesting that a longer duration of treatment with PTX is necessary to observe a therapeutic benefit on renal function.

Also in 2015, Chen et al. [78] published a retrospective analysis of 661 patients with CKD stages 3–5 treated with PTX. As in the PREDIAN trial, this study also explored the renoprotective effects of add-on PTX therapy to RAAS blockade. The authors observed that PTX provided nephroprotection in the subset of patients with higher levels of proteinuria, suggesting that proteinuria may be a predictor of response to PTX. Finally, Wu et al. [81] reported the results of an analysis of a nationwide administrative dataset of advanced CKD patients identifying two propensity score-matched cohorts: PTX users and nonusers. The authors found that the PTX group was protected from ESRD. This is the first evidence of the ability of PTX in reducing the risk of developing ESRD even in patients with advanced CKD.

## 5. Mechanisms Underlying the Renoprotective Effects of PTX

A meta-analysis published in 2008 pointed to the capacity of PTX to reduce the production of proinflammatory cytokines as the most likely explanation for its antiproteinuric action in patients with DKD [82]. A subsequent meta-analysis by Tian et al. [83] concluded that PTX therapy was also able to additively reduce proteinuria and urinary TNFα in patients with DKD under RAAS blockade. Importantly, this antiproteinuric action is observed without metabolic or hemodynamic changes [61].

PTX is a drug with several effects including the inhibition of phosphodiesterases (PDEs). The activity of PDEs modulates intracellular second messenger cyclic nucleotides levels in mammalian cells by controlling the degradation of cyclic adenosine-3,5-monophosphate (cAMP) as a feedback mechanism to return to basal levels [84,85]. The inhibition of PDEs by PTX prevents the inactivation of cAMP, resulting in increased levels of cAMP which in turn activate protein kinase A (PKA) [84], leading to a reduced synthesis of the pro-inflammatory cytokines IL1, IL6, and TNFα (Figure 1) [85,86]. Therefore, the inhibitory effects on PDEs exerted by PTX result in a reduction of inflammation.

PDEs are constituted by 11 gene-related families of isozymes (PDE1–PDE11) with more than 60 isoforms, and PTX is able to suppress the activity of several isoforms. In particular, PTX inhibits distinct cAMP hydrolyzing isozymes (i.e., PDE3 and PDE4) with the subsequent elevation of intracellular cAMP levels [87,88,89]. Importantly, PDE3 and PDE4 are mainly present in inflammatory cells [90]. The increase of cAMP secondary to the inhibition of PDE3 and PDE4 by PTX in turn activates PKA. In several models of renal disease, PTX is able to attenuate proteinuria via the modulation of signaling pathways or components triggered by inflammatory cytokines [91]. In a streptozotocin-induced diabetic rat model, PTX ameliorated sodium retention and renal hypertrophy together with a reduction in renal TNFα, IL1, and IL6 expression [39]. Similarly, in an alloxan-induced diabetic rat model, PTX also exerted anti-inflammatory effects via decreasing the levels of TNF α and IL6 [92].

Another favorable effect of PTX in the prevention or therapy of DKD could come from the impact on other factors directly related with kidney health. One of these factors is Klotho, a type I single-pass transmembrane protein predominantly expressed in the kidneys. Interestingly, Klotho is also found in blood and urine as a soluble form, which can be generated by cleavage of the transmembrane form or by secretion to the extracellular space of a shorter form derived from an alternative spliced transcript [93].

Klotho has been demonstrated to have beneficial biological effects including antiaging and nephroprotective functions. Some studies have reported diminished soluble Klotho levels in patients with type 2 DM [94,95], whereas a reduced renal Klotho expression has been observed in biopsies from patients with early stages of DKD [96], suggesting that soluble Klotho may be an early biomarker for predicting renal impairment in type 2 diabetic patients [97].

One of the most novel aspects regarding Klotho is the relationship with inflammation. The proinflammatory cytokines TNFα and TWEAK (tumor necrosis factor–like weak inducer of apoptosis) are able to produce a decrease in renal Klotho expression mediated by NF-κB, both in vitro and in vivo [98,99]. On the other hand, in renal cells and in human umbilical vein endothelial cells (HUVECs), the addition of Klotho inhibits the production of proinflammatory cytokines [99] and the expression of TNFα-induced adhesion molecules [100].

A recent post-hoc analysis of the PREDIAN trial by Navarro-Gonzalez et al. [79] has reported that the administration of PTX to type 2 diabetic patients with CKD stages 3 and 4 drives a reduction in serum and urinary TNFα as well as a significant increase in serum and urinary Klotho concentrations. Although the intimate mechanisms need to be further investigated, it may be suggested that PTX positively regulates Klotho based on its anti-inflammatory properties since changes in urinary TNFα were negatively and independently associated with the variation in serum and urinary Klotho concentrations. Moreover, in vitro experiments showed that PTX prevented the TWEAK and TNFα-induced Klotho downregulation in cultured renal tubular cells. In addition, the reduction of albuminuria in patients treated with PTX could contribute to this effect on Klotho expression, since albuminuria causes tubular inflammation and renal injury [101] and directly reduces Klotho levels in tubular cells in vivo and in culture [102].

## 6. Conclusions and Future Perspectives

Given the burden of diabetes worldwide, promising and relatively inexpensive therapeutic options for DKD derived from old drugs should not be overlooked. In this context, based on its anti-inflammatory effects, PTX may constitute a further therapeutic intervention in DKD patients to background RAAS blockade. Furthermore, PTX may have a beneficial impact on novel factors directly related with kidney health, such as Klotho. Although the intimate mechanisms need additional investigation, PTX may positively regulate Klotho on the basis of its anti-inflammatory properties. In addition to its potential repositioning for the treatment of DKD, PTX may also contribute to the design of novel therapeutic strategies aimed at preserving renal Klotho expression and therefore improving kidney and survival outcomes in CKD. Finally, despite the potential beneficial effects of PTX commented on in this review, it is necessary to recognize that at the present time there are no definitive data regarding renal outcomes. Therefore, large-scale multicenter trials, which are adequately powered, prospective, placebo-controlled, and with definitive endpoints on efficacy and safety, are necessary to demonstrate the renoprotective properties of PTF in the diabetic population with the maximum grade of evidence.

## Figures and Tables

**Figure 1 jcm-08-00287-f001:**
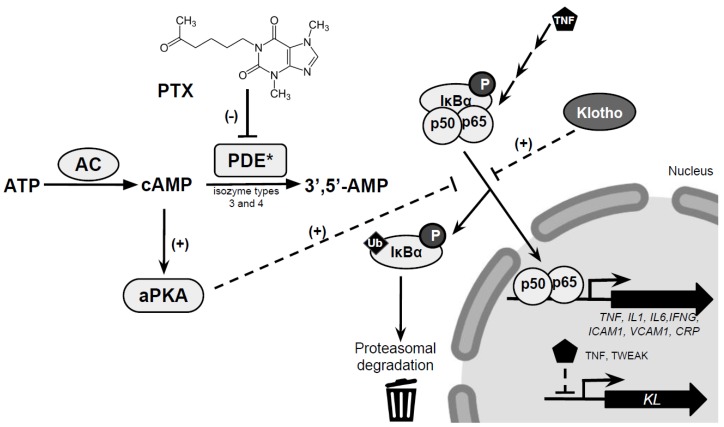
Suggested mechanisms of the anti-inflammatory effects of pentoxyfilline. Pentoxyfilline inhibits PDE activity increasing cAMP levels that activates PKA. Active PKA would inhibit ubiquitination that drives IκBα to 26S proteasome degradation and p50/p65 activation of the expression of citokynes and other genes. Decreased levels of TNF and TWEAK increases *KL* expression, whereas KL inhibits the production of pro-inflammatory cytokines and TNF-induced adhesion molecules. PTX; pentoxyfilline; PDE, phosphodiesterase; ATP, adenosine triphosphate; AC, adenylate cyclase; cAMP, cyclic adenosine-3,5-monophosphate; aPKA, active protein kinase A; IκBα, inhibitor of kappa B α; p50 (NF-κB1), nuclear factor NF-kappa-B p50 subunit (nuclear factor kappa-light-chain-enhancer of activated B cells 1); p65 (RelA), nuclear factor NF-kappa-B p65 subunit (V-Rel Avian Reticuloendotheliosis Viral Oncogene Homolog A); TNF, tumor necrosis factor α; IL, interleukin; IFNG, interferon gamma; ICAM1, intercellular adhesion molecule 1; VCAM1, vascular cell adhesion molecule 1; CRP, C reactive protein; TWEAK, TNF-related weak inducer of apoptosis; KL, Klotho.

**Table 1 jcm-08-00287-t001:** Examples of successfully repositioned drugs.

Drug	Original indication	Reposition
Amantadine	Influenza	Parkinson’s disease
Amphotericin	Antifungal	Leishmaniasis
Aspirin	Inflammation, pain	Antiplatelet
Bromocriptine	Parkinson’s disease	Diabetes mellitus
Bupropion	Depression	Smoking cessation
Colchicine	Gout	Recurrent pericarditis
Finasteride	Benign prostatic hyperplasia	Male pattern baldness
Gabapentin	Epilepsy	Neuropathic pain
Methotrexate	Cancer	Psoriasis, rheumatoid arthritis
Miltefosine	Cancer	Visceral leishmaniasis
Minoxidil	Hypertension	Male pattern baldness
Propranolol	Hypertension	Migraine prophylaxis
Sildenafil	Angina	Erectile dysfunction, pulmonary hypertension
Thalidomide	Morning sickness	Erythema nodosum leprosum
Zidovudine	Cancer	HIV/AIDS

**Table 2 jcm-08-00287-t002:** Main clinical studies on the use of PTX in diabetic nephropathy.

Ref.	Type of Study	Type of Intervention	Population	PTX Dose, Duration	Background RAAS Blockade	Main Findings	Anti-Inflammatory Effect
[65]	Randomized, controlled, open-label trial.	PTX vs. untreated	DM patients, *n* = 24 Albuminuria > 300 mg/24 h; creatinine clearance < 35 mL/min	400 mg/day, 6 months.	No.	59.3% proteinuria reduction in PTX-group (*p* < 0.001)	42.2% TNFα reduction in PTX-group (*p* < 0.001)
[66]	Randomized, controlled, open-label trial.	PTX vs. Captopril	DM patients, *n* = 39 Albuminuria > 300 mg/24 h; eGFR > 60 mL/min	1200 mg/day, 8 weeks	No.	PTX and Captopril reduced proteinuria; 40% in PTX-group (*p* < 0.05) and 38.5% in Captopril-group (*p* < 0.01)	Not reported
[67]	Randomized, controlled, open-label trial.	PTX vs. Captopril	DM patients, *n* = 130 UAE 20–200 μg/min.	1200 mg/day, 6 months.	No.	PTX and Captopril reduced proteinuria; 77.2% in PTX-group and 76.6 % in Captopril-group (*p* < 0.01 for both)	Not reported
[68]	Randomized, controlled, open-label trial.	PTX vs. untreated	DM patients, *n* = 61 Albuminuria > 300 mg/24 h; eGFR > 90 mL/min	1200 mg/day, 4 months.	ARB.	12.1% proteinuria reduction in PTX-group (*p* < 0.001)	28.1% and 28.8% reductions in serum and urinary TNFα, respectively (*p* < 0.01). TNFα changes were related to UAE
[69]	Randomized, double-blind controlled trial.	PTX vs. placebo	DM patients, *n* = 40 UAE 20–200 μg/min.	1200 mg/day, 4 months.	No.	73.8% and 84.6% reductions in urinary levels of both high and low molecular weight proteins (*p* < 0.05)	Not reported
[70]	Prospective trial	All in PTX	Patients with GN; non-diabetic, *n* = 17 Spot proteinuria > 1.5 g/g Cr; eGFR 24–115 mL/min/1.73 m^2^	800 mg/day, 6 months.	No.	36.5% and 33.9% reductions in spot and 24 h proteinuria (g/g Cr) (*p* < 0.01)	46% MCP-1 decrease (*p* < 0.01)
[71]	Prospective trial	All in PTX	CAN patients, *n* = 17 UAE 20–200 μg/min., mean eGFR 38 ± 8 mL/min/1.73 m^2^	1200 mg/day, 6 months.	No.	19.6% reduction of proteinuria at 3^rd^ month (*p* < 0.05) and improved graft survival	5.3% and 43.75% reductions in CD4+ cells bearing TNFα and IL10, respectively (*p* < 0.05)
[72]	Open-label, controlled trial	PTX vs. untreated	Diabetic glomerulosclerosis patients, *n* = 14 Proteinuria > 1.5 g/24 h; Cr clearance > 15 mL/min	400–800 mg/day, 1 year	ACEIs/ARBs.	PTX not reduced proteinuria or improved renal function	Not reported
[73]	Randomized, double-blind, controlled trial	PTX vs. placebo	Patients with GN, *n* = 18 proteinuria > 500 mg/24 h, mean eGFR 71.2 ± 30.6 mL/min/1.73 m^2^	800–1200 mg/day, 6 months.	ACEIs/ARBs.	56% reduction of proteinuria without affecting GFR	Not reported
[74]	Randomized, double-blind, controlled trial	PTX vs. placebo	CKD patients, *n* = 40 mean eGFR 29.5 ± 10.1 mL/min/1.73 m^2^, proteinuria greater than 1 g/24 h	800 mg/day, 1 year	ACEIs/ARBs.	PTX stabilized GFR. No reduction of proteinuria	Not reported
[75]	Randomized, controlled trial	PTX vs. untreated	CKD patients, *n* = 91 albuminuria > 300 mg/24 h, eGFR <60 mL/min/1.73 m^2^	800 mg/day, 1 year	ACEIs/ARBs.	PTX stabilized GFR. No reduction of proteinuria.	45.5 %, 11.1 %, and 57.4 % reductions in TNFα, fibrinogen and hsCRP, respectively (*p* < 0.05)
[76]	Randomized, controlled trial.	PTX vs. untreated	CKD patients, *n* = 56 Proteinuria > 500 mg/g of Cr; eGFR 10–60 mL/min/1.73 m^2^	400–800 mg/day, 1 year	ARB.	8.7% reduction of proteinuria compared to the control group (*p* < 0.001) stabilized GFR	Decrease in proteinuria was in conjunction with the decrease in TNFα and MCP1 (R = 0.64 and R = 0.55, respectively; *p* < 0.001 for both)
[77]	Randomized, controlled trial.	PTX vs. untreated	DM patients, *n* = 166 Albuminuria > 30 mg/24 h, eGFR 60–15 mL/min/1.73 m^2^	1200 mg/day, 2 years.	ARB.	Compared to the control group, 67.9% and 14.9% reduction in GFR decrease (*p* < 0.001) and proteinuria (*p* = 0.001) in the PTX-group, respectively.	10.6% reduction in urinary TNFα.
[78]	Single-center retrospective study	PTX vs. untreated	CKD patients, *n* = 661 Mean proteinuria 1102 mg/g of Cr, eGFR < 45 mL/min/1.73 m^2^	400–800 mg/day, 1 year.	ACEIs/ARBs.	PTX group showed a better renal outcome in patients with higher proteinuria (*p* = 0.005).	Not reported
[79]	Randomized, controlled trial. Post-hoc analysis.	PTX vs. untreated	DM patients, *n* = 166 Albuminuria > 30 mg/24 h, eGFR 60–15 mL/min/1.73 m^2^	1200 mg/day, 2 years.	ARB.	Compared to the control group, 5.9% and 9.3% increase in serum (*p* < 0.05) and urine Klotho (*p* < 0.001) in the PTX-group, respectively.	Changes in TNFα associated with changes of urinary Klotho (R^2^ = 0.60; *p* < 0.0001).

RAAS, Renin-Angiotensin Aldosterone System; ACEI, angiotensin converting enzyme inhibitor; ARB, angiotensin receptor blocker; CAN, chronic allograft nephropathy; CKD, chronic kidney disease; DM, diabetes mellitus; GN, glomerulonephritis; GFR, glomerular filtration rate; hsCRP, high sensitivity C reactive protein; MCP1, monocyte chemoattractant protein 1; PTX, pentoxifylline; TNFα, tumor necrosis factor α; UAE, urinary albumin excretion.

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
