# Peer review of "Pentoxifylline for Renal Protection in Diabetic Kidney Disease. A Model of Old Drugs for New Horizons"

_jcm, 2019, doi:10.3390/jcm8030287_

Reviewer 1 Report

Manuscript entitled “Pentoxifylline for Renal Protection in Diabetic Kidney Disease. A Case of Old Drugs for New Horizons” outlined importance of prevention of diabetic kidney disease, usefulness of drug repositioning, and potential of pentoxifylline that has an inhibitory effect of inflammation of patients with diabetic kidney disease. The authors showed simple figure of inhibitory mechanism of pentoxifylline in inflammation.

 The review includes important information of inhibitory effects of pentoxifylline against kidney disease in diabetic patients. The review is well written about effect of pentoxifylline to inhibit phosphodiesterase. In abstract section, the authors also described that pentoxifylline affects to Klotho. In conclusion section, the authors mainly discussed about effect of pentoxifylline to Klotho. However, effects of pentoxifylline against Klotho is difficult to understand in the present manuscript. Therefore, this manuscript should be improved about description of relationship between pentoxifylline and Klotho. I recommend that the authors make a figure, which is shown about molecular mechanisms between pentoxifylline and Klotho.

Author Response

Thank you very much for this comment. 

To date, our knowledge about the relationship between Klotho and inflammation is scarce. In fact, our study (Diabetes Care 2018) is the only one that has been published on this topic.

According to the reviewer’s suggestions, the final part of the manuscript (before the conclusion section) regarding the effect of pentoxifylline to Klotho has been revised in order to clarify it. Likewise, Figure 1 has been modified to show molecular mechanisms between pentoxifylline, inflammation and Klotho. Two new references have been included (#103 and #104).

Reviewer 2 Report

This paper reviews the interesting effects of pentoxifylline on proteinuria.

The authors should inform the reader regarding the number population size (number of subjects) in each trial they report in the table.

Furthermore, the authors should be more balanced in their opinion: pentoxifylline is not useful for peripheral arteropathy, as many trials have shown similar effects with placebo, and its use is not even recommended in current guidelines.

Therefore, the plausibility of pentoxifylline as nephroprotective agent might entirely depend on the size of previous works demonstrating nephroprotective effects rather than putative "vasoprotective" effects. Furthermore, the reduction of proteinuria is a proxy that could be misleading: many drugs that reduce proteinuria are indeed ineffective to counteract the progression of CKD.

Author Response

We thank so much the reviewer's comment.

The number population size (number of participants) in each trial in Table 2 has been included in the second column (Population).

 We agree with the reviewer about the need to be more cautious with the conclusions about the renoprotective effects of pentoxifylline. Therefore, the final part of the conclusion section has been modified to recognize this aspect.

Reviewer 3 Report

The article has been prepared correctly. It takes into account the current state of knowledge. The authors have taken into account important literature items. The work was written by experts who have been conducting research in this area for at least a few years (I rely on literature published in the Web of Sciences).
I think that this article need correct minor typos.

For example, in Figure 1 is "lsozyme" instead of "lisozyme".

Author Response

Thank you very much for your comments.

According to the reviewer's suggestion, the manuscript has been revised and several typos have been corrected.